# Structural Design of Dual-Type Thin-Film Thermopiles and Their Heat Flow Sensitivity Performance

**DOI:** 10.3390/mi14071458

**Published:** 2023-07-20

**Authors:** Hao Chen, Tao Liu, Nanming Feng, Yeming Shi, Zigang Zhou, Bo Dai

**Affiliations:** 1The State Key Laboratory of Environment-Friendly Energy Materials, Southwest University of Science and Technology, Mianyang 621010, China; chenhaoswustedu@163.com (H.C.); 18780550401@163.com (N.F.); seiyem@outlook.com (Y.S.); 2School of Materials and Chemistry, Southwest University of Science and Technology, Mianyang 621010, China; 3School of Mathematics and Science, Southwest University of Science and Technology, Mianyang 621010, China; liut427619@163.com (T.L.); zhouzigang@swust.edu.cn (Z.Z.)

**Keywords:** thin-film type, heat flow sensor, thermal resistance layer, thermopile, finite element simulation

## Abstract

Aiming at the shortcomings of the traditional engineering experience in designing thin-film heat flow meters, such as low precision and long iteration time, the finite element analysis model of thin-film heat flow meters is established based on finite element simulation methods, and a double-type thin-film heat flow sensor based on a copper/concentrate thermopile is made. The influence of the position of the thermal resistance layer, heat flux density and thickness of the thermal resistance layer on the temperature gradient of the hot and cold ends of the heat flow sensor were comprehensively analyzed by using a simulation method. When the applied heat flux density is 50 kW/m^2^ and the thermal resistance layer is located above and below the thermopile, respectively, the temperature difference between the hot junction and the cold junction is basically the same, but comparing the two, the thermal resistance layer located above is more suitable for rapid measurements of heat flux at high temperatures. In addition, the temperature difference between the hot and cold contacts of the thin-film heat flux sensor increases linearly with the thickness of the thermal resistance layer. Finally, we experimentally tested the response–recovery characteristics of the sensors, with a noise of 2.1 μV and a maximum voltage output of 15 μV in a room temperature environment, respectively, with a response time of about 2 s and a recovery time of about 3 s. Therefore, the device we designed has the characteristic of double-sided use, which can greatly expand the scope of use and service life of the device and promote the development of a new type of heat flow meter, which will provide a new method for the measurement of heat flow density in the complex environment on the surface of the aero-engine.

## 1. Introduction

With the rapid development of China’s aerospace industry, the design and development of aero-engines, as the core components of aero-aircraft, must rely on advanced and effective testing technologies to verify their design performance indicators and reliability [1,2]. As the most important component of an aero-engine, the turbine blades have a narrow internal space and a harsh working environment, so it is necessary to accurately measure the surface temperature distribution and heat transfer process of high-temperature components such as turbines and combustion chambers to verify the cooling efficiency and performance of the thermal barrier coating, evaluate the combustion efficiency of the gas, monitor the engine running state, and troubleshoot [3,4,5]. Therefore, rapid and accurate measurement of the heat distribution and transfer characteristics on the surface of turbine blades of aero-engines is essential for the development of the aerospace industry. 

A heat flux sensor is a kind of device or instrument that can detect and monitor the heat flux density in real time and convert it into an electrical signal for output [6,7,8]. It is widely used in the technical field of national defense, and the technology of measuring heat flux density has made great progress. In order to describe the harsh environmental changes of high-temperature airflow and high-pressure airflow on the turbine engine blades of the space shuttle more thoroughly, in addition to the temperature on the turbine engine blades as a parameter, heat flux density, as another important measurement parameter, has gradually begun gain interest the aviation technical field [9,10]. In order to solve the limitation of the heat flow sensor in high temperature and high pressure air flow environment, researchers have conducted extensive theoretical research and experimental verification. Traditional heat flow meters such as the Gordon heat flow meter [11], the Schmidt–Belter meter [12] and the standard stratification meter [13] are unable to meet the test requirements, such as being large, affecting the accuracy of heat flow measurements and being unsuitable in fields such as aerospace where a high measurement accuracy is required [14,15,16,17]. As a result, thin-film heat flow meters only began to appear in aerospace engine research in the 1990s, measuring a wide range of heat flows with a large output signal, making them one of the most widely used devices for measuring heat flow density today. Combined with MEMS (micro electro mechanical system) technology [18,19,20], the thin-film sensor is deposited on the surface of the measured object, which is integrated with the measured object, and has the characteristics of small volume, small heat capacity, low interference and no damage to the air flow on the surface of the component, which will also provide a new measurement method for the surface heat flow measurement of engine hot-end components [21,22].

In this paper, the finite element method is used to design and optimize the structure and position of the thermal resistance layer, and a MEMS thin-film heat flow sensor is prepared. The sensor is made of a copper/concentrate thin-film thermopile as the sensing structure, and SiO_2_/Al_2_O_3_ as the bottom thermal resistance layer. The effect of the thermal resistance layer position, heat flow density and thickness of the thermal resistance layer on the temperature gradient of the cold and hot junction of the thermal flow sensor is investigated and the simulation results are compared with the experimental measurement results. The results of the simulations are compared with the experimental results to provide theoretical guidance and experimental validation for the further validation and development of a high-performance and stable heat flow sensor for monitoring heat flow density in the complex environment of aero-engine surfaces.

## 2. Working Principle

The working principle of a thin-film thermopile heat flow meter can be explained by heat conduction theory, as shown in Figure 1; that is, when there is heat flow through the sensor, the temperature changes only in the X direction, resulting in a temperature gradient. According to Fourier’s law, the heat flow density through the heat flow meter can be obtained.

The heat flow density Q through the heat flow meter is expressed according to Equation (1) [22,23]:(1)$$Q=-\lambda \frac{\Delta T}{\Delta X}=-\lambda \frac{T_{2}-T_{1}}{h}$$
where Q is the heat flow density, *λ* is the thermal conductivity of the thermal resistance layer, T_1_ and T_2_ are the temperatures of the upper and lower surfaces of the thermal resistance layer, respectively, and *h* is the thickness of the thermal resistance layer.

According to Equation (1), if the material and geometry of the device are determined, the magnitude of the heat flow density can also be obtained by measuring this temperature difference. And, according to the thermopile temperature measurement principle [24,25]:(2)E=S⋅N⋅ΔT
where E is the magnitude of the voltage output, S is the Seebeck coefficient of the thermocouple and N is the number of pairs of thermocouples at the base of the device. Combining Equation (1) with Equation (2) yields the heat flow density versus thermopile output as Equation (3):(3)$$Q=\frac{\lambda }{S\cdot N\cdot h}E$$

With the above analysis, we can obtain the heat flux density by knowing the temperature difference of each thermocouple pair at the bottom or directly measuring the thermal potential.

## 3. Sensor Structure Design

As shown in Figure 2, according to the heat conduction theory and thermoelectric effect, the structural design of the dual-type thin-film heat flow sensor is schematically shown in Figure 2a, where the top structure is mainly composed of alternating Al_2_O_3_ and SiO_2_ thermomechanical layers. Figure 2b shows the 3D structural diagram of the bottom view of the device, and below the thermal resistance layer is the thermopile layer (copper/concentrate thermopile), which is mainly used to convert the temperature difference on the lower surface of the resistance layer into a voltage for detection. It is worth noting that we have designed 200 pairs of thermocouples. This is because the voltage provided by a single pair of thermocouples is too small, so we have designed a reasonable structure to connect 200 pairs of thermocouples in series, which greatly increases the voltage output and reduces the influence of some accidental factors. The geometric parameters of the finite element model are shown in Table 1. In order to explain the working principle of the new heat flow meter we designed more clearly, we drew a side view of the device. When the heat flow is applied vertically to the sensor, due to the difference in thermal conductivity between the SiO_2_ and Al_2_O_3_ thermal resistive layers, the Al_2_O_3_ thermal resistive layer with high thermal conductivity transfers the excess heat from the cold end to the Al_2_O_3_ ceramic substrate to form the cold end, while the SiO_2_ thermal resistive layer with low thermal conductivity restricts the heat transfer from the hot end to the substrate to form the hot end, which creates a temperature gradient between the hot end and the cold end. The underlying temperature distribution of the bottom layer is also different, so we detect the temperature distribution of the bottom layer by adding a thermopile layer [26,27,28]. Therefore, it is possible to infer the heat flow in the top layer or the power of the laser by observing the output voltage.

## 4. Finite Element Simulation Analysis

### 4.1. Effect of Different Thermal Resistance Layer Positions on Heat Flow Measurements

In order to ensure that the designed sensor can show good response characteristics, the finite element software of COMSOL version 5.4 was used for simulation analysis, a boundary heat source was added to simulate the heat flow, and the connection position between thermocouple and thermal resistance layers is meshed to obtain the theoretical response characteristics and sensitivity of the sensor and compared with experimental results. The structural parameters of the finite element model created by the simulation are the same as those of the prepared device. Figure 3a shows the temperature distribution curve of the thin-film heat flow sensor after reaching steady state when the thermal resistance layer is located above the thermopile design structure and the heat flow of 50 KW/m^2^ is vertically applied to the sensor. Figure 3b shows the temperature cloud on the sensor surface. It can be clearly observed that due to the difference in thermal conductivity between the SiO_2_ and Al_2_O_3_ thermal resistance layers, a temperature difference is generated on the thermopile that leads to a voltage output, which indicates that the design of the structure of the thin-film heat flow sensor is reasonable. It can also be seen that the temperature below Al_2_O_3_ layer is higher, with an average temperature at the hot end of 298.704 K, and the temperature below SiO_2_ layer is lower, with an average temperature at the cold end of 298.663 K and a temperature difference of 0.0409 K. It is worth mentioning that the average temperature of the hot end when the thermal resistor layer is located above the thermopile is lower than that when it is located below, which is more suitable for rapid measurement of heat flow at low temperatures.

Figure 3c shows the temperature distribution curve of the thin-film heat flow sensor when a heat flow of 50 KW/m^2^ is applied vertically to the sensor after reaching steady state when the heat resistance layer is located below the thermopile design structure, and Figure 3d shows the temperature cloud on the surface of the sensor. It can be seen that the temperature at the hot end of the thermocouple is roughly 298.748 K, while the average temperature at the cold end is 298.707 K, with a temperature difference of 0.041 K.

According to the Seebeck effect, the output voltage when the thermal resistance layer is located below the thermopile design structure is 13.1 μV. This is because the carriers within the conductor move from the hot end to the cold end under the temperature gradient, and accumulate at the cold end, thus generating a potential difference inside the material, and under this potential difference, a reverse charge flow is generated. When the thermal charge flow and the internal electric field are in dynamic balance, a stable temperature difference potential is formed at both ends of the material. The smaller the thickness of the thermal resistance layer, the shorter the heat transfer time and the faster the response time.

### 4.2. Effect of Different Thermal Resistance Layer Thicknesses on Heat Flow Measurements

The thickness of the thermal resistance layer of a thermoelectric stack-type heat flow sensor has a huge impact on its performance. The greater the thickness, the longer the heat transfer time, and the greater the change of the temperature distribution on the lower surface. In order to study the influence of thermal resistance layers with different thicknesses (3, 4, 5, 6, 7 and 8 μm) on the performance, simulations were carried out. With a steady-state heat flow of 50 KW/m^2^ applied, Figure 4a shows the temperature distribution profile of the lower surface of the thermal resistance layer at different thicknesses. The temperature at the hot end remains almost constant at around 298.704 K, but the temperature at the cold end increases with increasing thickness of the thermal resistance layer. Figure 4b shows the pattern of temperature difference between the hot and cold junctions with the thickness of the thermal resistance layer. When the thickness of the thermal resistance layer is 3 μm, the temperature difference between the hot and cold junctions is 0.008 K, but when the thickness is increased to 8 μm, the temperature difference increases to 0.16 K. As the thickness of the thermal resistance layer increases, the temperature at the hot end of the thin-film heat flow sensor remains constant, the temperature at the cold end increases linearly and the temperature difference between the hot and cold junctions also increases linearly, indicating that by increasing the thickness of the thermal resistance layer, the sensitivity of the device can be effectively improved. But, at the same time, the response time requirement must also be considered. Therefore, when choosing the thickness of the thermal resistance layer, we need to consider the balance between sensitivity and response time, because as the thickness of the thermal resistance layer increases, the heat transfer time and response time will be prolonged [29,30]. In a word, the heat flow sensitivity of the sensor can be effectively improved by reducing the thickness of the thermal resistance layer. However, we should also note that in order to meet the specific requirements and optimize the performance of the sensor, it is necessary to consider the corresponding reduction in heat transfer time, equipment heat transfer protection and response time requirements in combination with these factors.

### 4.3. Effect of Different Heat Flow Densities on Heat Flow Measurements

The devices we designed are mainly used to detect the magnitude of heat flow density or laser power density, so we investigated the change in the temperature of the bottom layer by varying the Q value applied to the top layer of the thermal resistive layer [31]. Figure 5 shows the temperature distribution on the sensor surface when the continuous heat flow of 10, 50, 100, 150 and 200 KW/m^2^ acts on the sensor. As can be seen in Figure 5a, the overall temperature distribution of the device increases with increasing Q as the Q value varies from 10 to 200 KW/m^2^. When Q = 100 KW/m^2^, the hot end temperature is 304.258 K, and the cold end temperature is 304.176 K, so the stable temperature difference between the hot and cold end is 0.082 K. When Q = 100 KW/m^2^, the temperature difference between the hot and cold end is ∆T = 0.163 K. Therefore, as shown in Figure 5b, the device Q value versus the corresponding ∆T, in which the corresponding ∆T increases as the i Q value increases, shows a good linear relation. Therefore, the thin-film heat flux sensor structure has a good output in the simulation test and can realize the accurate measurement of heat flux density. According to our simulation results, it is shown that our thin-film thermal flow sensor has good output precision and accuracy in measuring the heat flux. In a word, our research shows that by detecting the temperature change of the bottom layer, the accurate measurement of the heat flux can be realized. Our thin-film heat flux sensor structure shows good performance in simulation tests and can provide accurate heat flux measurements. 

## 5. Fabrication and Performance Testing of Heat Flow Sensors

### 5.1. Heat Flow Sensor Production

Figure 6 shows the flow chart of the physical preparation of the designed heat flow meter. The main four coating processes were used to complete the fabrication [30,32], and the specific operation steps are shown in Figure 7: ① prepare the 4-inch silicon wafer for ultrasonic cleaning with deionized water, acetone and anhydrous ethanol for 15 min each; ② the first coating material is Al_2_O_3_, and then the final Al_2_O_3_ thermal resistance layer is formed through glue dumping, exposure, development and etching; ③ the second coating material is SiO_2_, and the final SiO_2_ layer is formed through glue dumping, exposure, development, ion etching and acetone debinding; ④ the above two coating processes are repeated to form the thermopile layer; and ⑤ finally, the metal electrodes are etched and soldered. The prepared sensor is shown in Figure 8, where the structural dimensions of the sensor are shown in Figure 8a,b, which show an optical microscope view of several interconnected thermopile strips. The thermopile layer (copper and con-copper layer) mainly uses the glow discharge principle to make argon gas ionization under high pressure after the formation of positive ions and bombardment of the target surface, so that the target particles sputtered out to reach the surface of the substrate to form a thermopile film of copper and con copper [33,34].

### 5.2. Performance Testing of Heat Flow Sensors

As shown in Figure 9a, this study builds a high-temperature steady state and transient heat flow calibration system and conducts relevant experiments through the calibration platform and an infrared lamp to measure the sensitivity and reliability of the sensors, and then effectively evaluate the performance of the new thin-film-type thermo-fluid mete. Here we have chosen quartz infrared lamps as a highly efficient heating source, which has a quartz bubble shell and tungsten filament composition to avoid the blackening of the tube wall and the consequent decrease in light flux. Advantages of quartz infrared lamps include: high efficiency, fast heat transfer, and fast responses. Figure 9b shows the dynamic response curves of the prepared sensor samples in terms of the heat flow density of the IR lamp radiation. It can be seen that the sample exhibits excellent response–recovery characteristics with a noise of 2.1 μV and a maximum voltage output of 15 μV at room temperature, respectively, with a response time of about 2 s and a recovery time of about 3 s. This is in good agreement with the above simulation results. Wherein, the response time is the time from the exposure of the thermofluid meter to a heat flow of a given power to the output stabilizing the output voltage under test conditions; typically, the time to reach 90% of the maximum value is read as the response time. Whereas, the recovery time is the time from when the thermofluid meter is no longer in contact with the heat flow to when the output returns to a steady state under test conditions; typically, the time to recover to 10% of the maximum value is read as the recovery time [35,36]. In addition, the magnitude of the output voltage shows good agreement with the simulation calculations.

Figure 10 shows the static response curves of the device under different heat flow densities, and it can be seen that the heat flow densities are in the range of 0–160 kW/m^2^, and the output voltages are 0.13 mV, 0.27 mV, 0.27 mV, 0.39 mV, 0.93 mV, 2.25 mV, and 3.58 mV, respectively, when the heat flow densities are, in order, 20 kW/m^2^, 25 kW/m^2^, 30 kW/m^2^, 50 kW/m^2^, 100 kW/m^2^, and 150 kW/m^2^, and the corresponding output voltages are 0.39 mV, 0.93 mV, 2.25 mV and 3.58 mV, indicating that the sensor output thermoelectric potential shows a good linear relationship with heat flow. The test results were linearly fitted to obtain the binary sublinear equation, y = 0.02648x − 0.39679, the slope of which is the sensor sensitivity, which is 0.02648 mV/(kW/m^2^).

## 6. Conclusions

In this paper, a double-structure thin-film heat flow sensor was designed, and a finite element model is established. The output voltage is adjusted by the relationship between the position of the thermal resistance layer and the thermopile, and the temperature difference between the hot end and cold ends of the heat flow sensor remains basically constant under the condition of the applied heat flow density of 50 kW/m^2^. As the thickness of the thermoresistive layer increases, the cold-hot junction temperature difference of the thin-film thermofluid sensor increases linearly, from 0.008 K to 0.163 K, as the thickness of the thermoresistive layer increases from 3 μm to 8 μm. In addition, the results of the finite element simulation are compared with the experimental results. The results show that the sample has good response–recovery characteristics, the noise is 2.1 μV at room temperature, the maximum voltage output is 15 μV, the response time is about 2 s, and the recovery time is about 3 s, which is in good agreement with the above simulation results. The preparation process has good repeatability, which provides good technical support for the monitoring of heat flux density, standardization and mass production of aero-engine surface in a complex environment.

## Figures and Tables

**Figure 1 micromachines-14-01458-f001:**
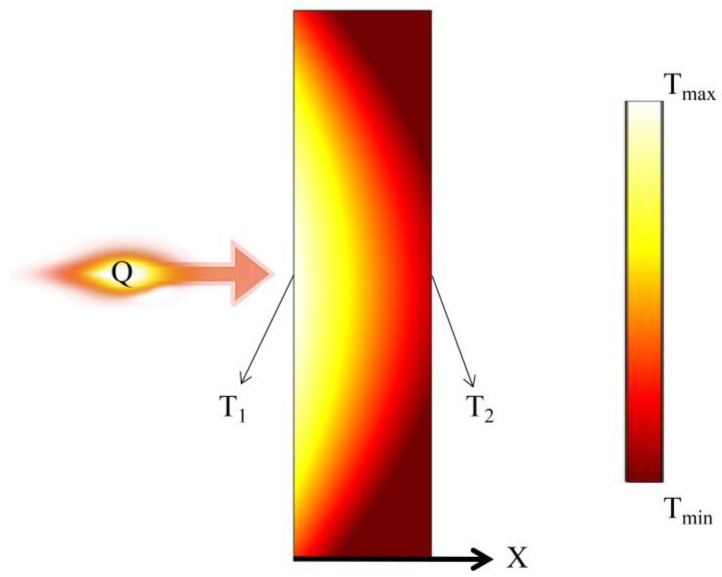
Schematic diagram of Fu Liye’s law.

**Figure 2 micromachines-14-01458-f002:**
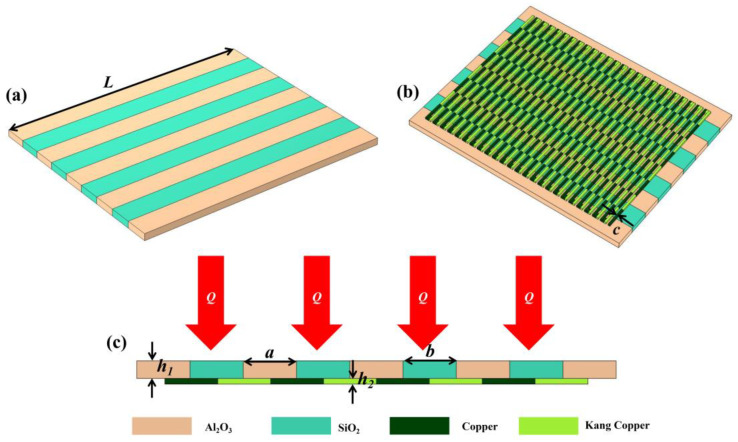
Structural diagram of the heat flow meter. (**a**) Front three-dimensional structure drawing. (**b**) Reverse three-dimensional structure. (**c**) Side view of the heat flow meter.

**Figure 3 micromachines-14-01458-f003:**
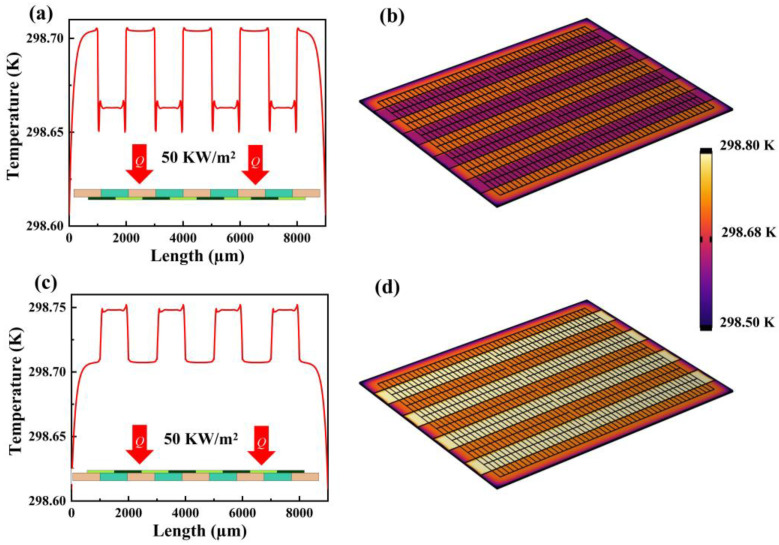
Temperature distribution profile when applying a heat flow density of 50 KW/m^2^. (**a**) Temperature distribution profile on the thermopile layer for frontal incidence of heat flow. (**b**) Temperature cloud over the thermopile layer at frontal incidence. (**c**) Temperature profile on the thermopile layer at frontal incidence. (**d**) Temperature cloud over the thermopile layer at frontal incidence.

**Figure 4 micromachines-14-01458-f004:**
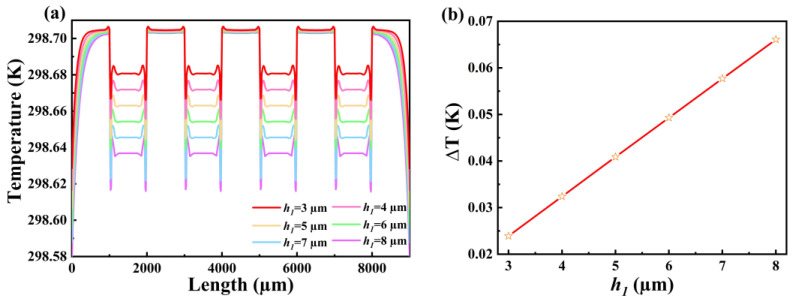
Variation of (**a**) temperature distribution curve and (**b**) temperature difference of individual thermopiles for thermopile layers with different thicknesses of thermal resistance layer.

**Figure 5 micromachines-14-01458-f005:**
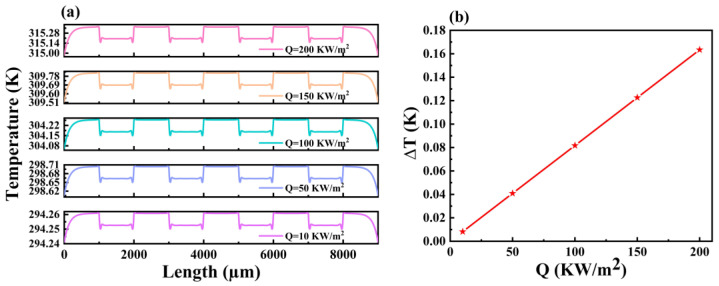
Temperature distribution profiles on (**a**) the thermopile layers and (**b**) the temperature difference (∆T) variation on individual thermopiles when different heat flow densities are applied.

**Figure 6 micromachines-14-01458-f006:**
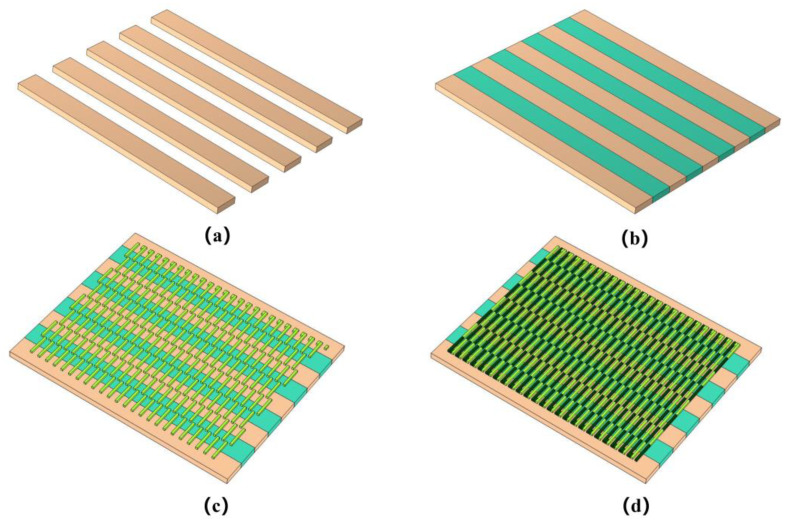
Schematic diagram of the preparation process. (**a**) Deposition of the Al_2_O_3_ thermal resistance layer. (**b**) Deposition of SiO_2_ thermal resistance layer. (**c**) Deposition of the copper-containing layer of the thermopile. (**d**) Deposition of the copper layer of the thermopile.

**Figure 7 micromachines-14-01458-f007:**
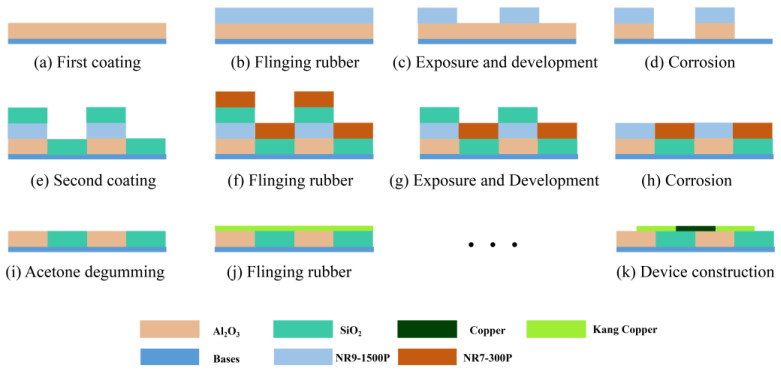
Process design flow diagram for the preparation of thin-film thermopiles.

**Figure 8 micromachines-14-01458-f008:**
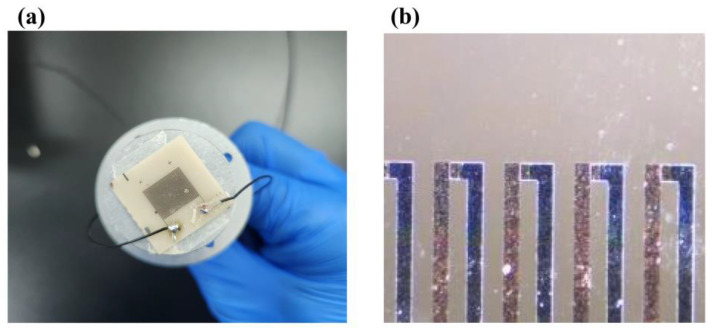
(**a**) Physical drawing of the structural dimensions of the sensor. (**b**) An optical microscope view of the thermopile.

**Figure 9 micromachines-14-01458-f009:**
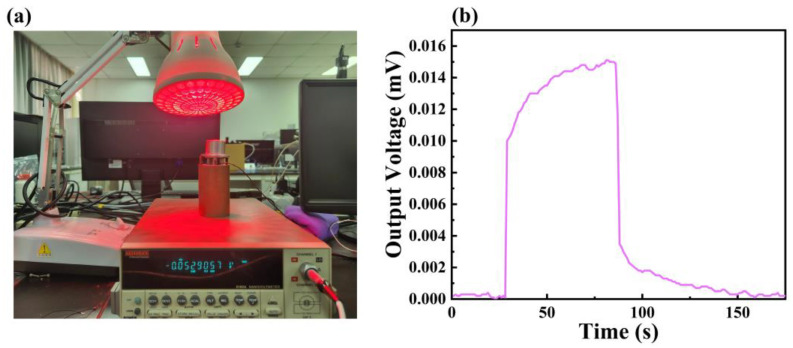
(**a**) Physical diagram of the experimental test setup. (**b**) Output voltage–time dynamic variation curve of the sample under IR lamp radiation.

**Figure 10 micromachines-14-01458-f010:**
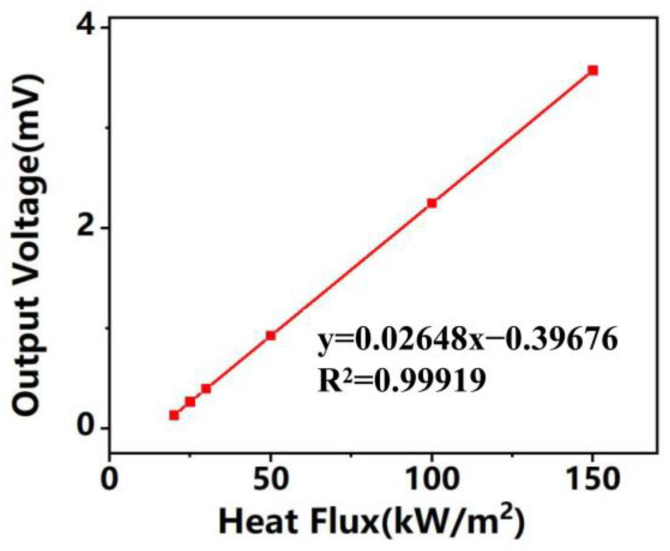
Steady-state calibration measurements of a dual-type thin-film heat flow meter.

**Table 1 micromachines-14-01458-t001:** Dual type thin film thermopile heat flow sensor structure parameter table.

Parameter Name	Description of Parameters	Parameter Values
*Q*	Heat flow density	50 KW/m^2^
*h_1_*	Thickness of thermal resistance layer	5 μm
*h_2_*	Thermoelectric stack thickness	1 μm
*L*	Length of thermal resistance layer	1 μm
*a*	Width of Al_2_O_3_	1000 μm
*b*	Width of SiO_2_	1000 μm
*c*	Thermocouple length	100 μm
N	Number of thermocouples	200

## Data Availability

Data underlying the results presented in this paper are not publicly available at this time but may be obtained from the authors upon reasonable request.

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
