# Peer review of "Structural Design of Dual-Type Thin-Film Thermopiles and Their Heat Flow Sensitivity Performance"

_micromachines, 2023, doi:10.3390/mi14071458_

Round 1

Reviewer 1 Report

In this work, the authors reported a double-type thin-film heat flow sensor based on a copper/concentrate thermopile. The finite element simulation was performed to illustrate the effect of thickness on the heat flow measurements. And they also fabricated a sensor based on the simulation results to characterize the sensor performance. However, the novelty behind this work was not well clarified and the experimental results were not detailed. Before further consideration, I still have several comments and suggestions.

1. I suggest the authors simplify the abstract to highlight the innovation and important results of this work.

2. What is the detailed coating method of SiO2 and Al2O3? How did the authors control the thickness of these layers?

3. How did the authors define the recovery and response time? From Figure 9b, it was obviously the recover time could be much longer than response time?

4. What was the condition for the best performance shown in experimental results? For example, the thickness of SiO2 and Al2O3 layer and the value of applied heat flow density?

5. The authors claimed that the experimental result was in good agreement with the above simulation results. However, it was difficult to compare the experimental results with the simulation results because the authors did not give the detailed information of the as-fabricated device and some important parameters were not given in the simulation results such as response and recovery time. I suggest the authors give a table to carefully compare the parameters of the experimental results with the simulation ones under the same condition.

6. What is the sensor sensitivity of the as-fabricated sensors?

7. The novelty behind this work was not clear. Is it a novel structure? Or is the performance better than other sensors? Please comment it.

8. From the Introduction, the main challenge for heat flow sensor is to stably work in harsh environment such as high temperature and high pressure. However, no evidence or experimental results were demonstrated in the as-fabricated device to demonstrate its potential for application in such harsh environment.

Reviewer 2 Report

The authors have made significant contributions by developing a compelling double-type thin-film heat flow sensor for aerospace engines, addressing design limitations. They employed a copper/concentrate thermopile and finite element analysis to investigate various factors' impact on temperature gradients. The results demonstrate consistent sensor performance at specific heat flux densities. Both experimental and simulation approaches were used to validate the outcomes. The manuscript is well-written and engaging, making it accessible to a broad readership. It serves as a valuable resource for the aerospace industry and researchers working on sensing applications.

To further improve the manuscript, the authors are recommended to  consider addressing the following comments:

 1)      The authors should provide a clear explanation for the discrepancy between the experimental output voltage time dynamics and the theoretical investigations, particularly in terms of response and recovery times (as shown in Figure 9b). The observed recovery time of approximately 60 seconds contrasts with the expected response and recovery times of 2-3 seconds.

 2)      To provide a more comprehensive understanding of the sensor's behavior and validate the repeatability of the experimental findings (as shown in Figure 9b), authors are advised to include data for multiple continuous cycles instead of presenting results from a single cycle. This will enhance the reliability of their observations. Additionally, for better accuracy of thermal sensing (response and recovery), authors might consider using a heat gun instead of an IR lamp. This alternative method may provide better accuracy as it would help mitigate the presence of residual heated atmosphere within the area even after turning off the IR lamp.

 3)      The authors should elaborate on the abrupt temperature changes observed at the interface of Al2O3 and SiO2, which lead to sharp peaks or dips (as seen in Figure 3a). Providing an explanation for this phenomenon and incorporating it into the main manuscript would enhance the discussion.

 4)      Consider rewriting the abstract by breaking down lengthy sentences into smaller ones, improving readability and making it more accessible to readers.

 5)      Please correct the typographical error in line #21 (a missing space between "recovery" and "characteristics") also need to check some other potential errors throughout the manuscript.

NA

Round 2

Reviewer 1 Report

The authors have carefully revised the manuscript according to the reviewer's comments. I suggest its publication at present form.